

# Two new species and the molecular phylogeography of the freshwater crab genus *Bottapotamon* (Crustacea: Decapoda: Brachyura: Potamidae)

Ning Gao, Ying-Yi Cui, Song-Bo Wang and Jie-Xin Zou

Research Laboratory of Freshwater Crustacean Decapoda & Paragonimus, School of Basic Medical Sciences, Nanchang University, Nanchang, Jiangxi Province, China

## ABSTRACT

*Bottapotamon chenzhouense* sp. n. and *B. luxiense* sp. n. are described from Hunan Province and Jiangxi Province, respectively. These species both have diagnostic features of the genus *Bottapotamon* and discernible characteristics as new species. *B. chenzhouense* sp. n. can be distinguished from co-geners by features such as the G1, which has a fold covering the surface of the entire subterminal article with a distal region. *B. luxiense* sp. n. has an elliptical carapace, and a sturdy and blunt terminal article of G1. The molecular phylogeny and biogeography of the genus *Bottapotamon* (Decapoda: Brachyura: Potamidae) were studied, using mitochondrial cytochrome oxidase I (mtDNA COI), 16S rRNA and nuclear histone H3 gene fragments. The results support the assignment of the two new species to the genus *Bottapotamon*. In addition, the divergence time of the genus *Bottapotamon* was estimated to be 3.49–1.08 Ma, which coincided with various vicariant and dispersal events that occurred in the geological area where the genus *Bottapotamon* is commonly distributed. Mountains appear to have played an important role in the distribution of this genus. The Wuyi Mountains gradually formed offshore and inland of southeastern China by the compression of the Pacific plate and the Indian plate in the Neogene-Quaternary, and the Luoxiao Mountains formed continuously in the continued forming in the north-south direction because of neotectonic movement, have resulted in the geographical distribution pattern of the genus *Bottapotamon*, which was also established gradually.

## INTRODUCTION

The genus *Bottapotamon* is a unique genus of freshwater crabs from the China mainland. In 1997, three species of the genus *Malayopotamon* on (*Bott, 1967*; *Cheng, Lin & Luo, 1993*; *Dai et al., 1979*) and one new species were identified as *Bottapotamon* on the basis of its morphological characteristics, such as the form of carapace and first gonopod (G1) (*Türkay & Dai, 1997*). Until the current study, the genus *Bottapotamon* contained *B. fukiense* (*Dai et al., 1979*), *B. engelhardti* (*Bott, 1967*), *B. yonganense* (*Cheng, Lin & Luo,*

Corresponding author
Jie-Xin Zou, jxzou@ncu.edu.cn

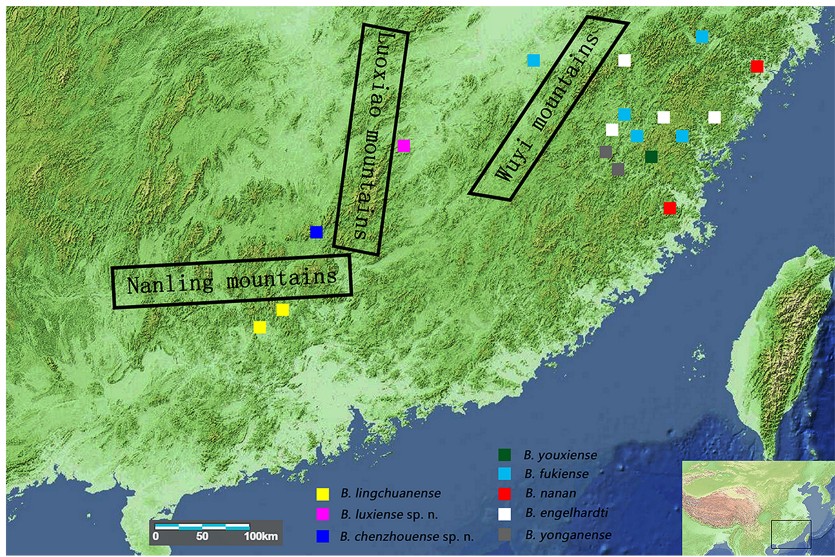

**Figure 1  Collection sites for the genus *Bottapotamon*.** The regional map comes from https://commons.wikimedia.org/wiki/Atlas_of_the_world and http://landsatlook.usgs.gov/; the map was edited with Adobe Photoshop CS6.

*1993*), *B. lingchuanense* (*Türkay & Dai, 1997*), *B. youxiense* (*Cheng, Lin & Li, 2010*) and *B. nanan* (*Zhou, Zhu & Naruse, 2008*).

The relatively low fecundity and poor dispersal abilities of freshwater crabs (*Daniels et al., 2003*; *Yeo et al., 2008*) mean that these crabs are easily isolated by barriers such as mountains or seas. Geographically isolated populations then become genetically natural distinct and result in allopatric speciation (*Shih et al., 2006*; *Yeo et al., 2007*). In mainland China, the distribution of the genus *Bottapotamon* is restricted within the area of the Wuyi Mountain Range; *B. engelhardti*, *B. yonganense*, *B. youxiense* and *B. nanan* are distributed east of the Wuyi Mountain Range, *B. fukiense* occurs on both sides of the Wuyi Mountains (Fujian and Jiangxi Provinces), and only *B. lingchuanense* has been isolated in the Nanling Mountain Range (*Dai, 1999*) (Fig. 1). The geographic barrier separating the Wuyi Mountains from the Nanling Mountains is the Luoxiao Mountain Range, which is the highest range in the area, exceeding 2,120 m in height (*Gong, Zhuang & Liao, 2016*). The terrain the genus *Bottapotamon* now inhabits is geologically relatively stable and experienced little orogenic activity during the Cenozoic Era (*Yi, 1996*; *Zhou & Li, 2000*). Therefore, we hypothesize that the current distribution of the genus *Bottapotamon* in mainland China was caused by the emergence of these mountains.

While organizing the existing specimens deposited at the Department of Parasitology of the Medical College of Nanchang University (NCU MCP) and the newly collected specimens, the first and third author discovered two new species collected from Chenzhou City, Hunan Province, and Luxi County, Jiangxi Province, respectively. This paper compares the morphological features of eight species including two new species of the genus *Bottapotamon*, as well as 16S rRNA (*Crandall, Fitzpatrick & Faith, 1996*), mtDNA

COI (*Folmer et al., 1994*) and nuclear histone H3 (*Colgan et al., 1998*) gene fragments that are uesd to support the establishment of new species in the genus *Bottapotamon.* The phylogenetic relationship, distribution pattern and possible association with major geological and historical events are also discussed.

## MATERIALS & METHODS

### Specimens collection

Specimens from Jiangxi, Zhejiang, Fujian and Guangxi, were recently collected and preserved in 95% ethanol. The remaining specimens used in this study were from and deposited at the Department of Parasitology of the Medical College of Nanchang University (NCU MCP), Jiangxi Province, China. The authors compared specimens with holotypes of the National Zoological Museum of China, Chinese Academy of Sciences (CAS). All 26 specimens were used for mtDNA COI, 16S rRNA and histone H3 gene fragment amplification (Table 1).

### Phylogenetic analyses and Divergence time estimation

Genomic DNA was extracted from leg muscle tissue with an OMEGA EZNA$^{TM}$ Mollusc DNA Kit. The 16S rRNA, mtDNA COI, and histone H3 regions were selected for amplification by polymerase chain reaction (PCR) (Table 2). The amplification products were sent to the Beijing Genomics Institute for bidirectional sequencing, and the sequencing results were spliced manually to obtain the sequence data. DNA sequences of *B. yonganense* specimens collected from the suburb of Sanming City, Fujian Province, China, could not be amplified due to poor preservation.

The sequences of four individuals with the same primer sequences were selected from National Center for Biotechnology Information (NCBI) database, as the outgroups (*Candidiopotamon rathbunae* (GenBank accession numbers: mtDNA COI—AB290649, 16S rRNA—AB208609, histone H3—AB290668), *Geothelphusa dehaani* (GenBank accession numbers: mtDNA COI—AB290648, 16S rRNA—AB290630, histone H3—AB290667), *Himalayapotamon atkinsonianum* (GenBank accession numbers: mtDNA COI—AB290651, 16S rRNA—AB290632, histone H3—AB290670), and *Ryukyum yaeyamense* (GenBank accession numbers: mtDNA COI—AB290650, 16S rRNA—AB290631, histone H3—AB290669). After comparing and selecting the conservative regions, each sequence was 1323 bp in length. According to the Akaike information criterion (AIC), MrMTGui: ModelTest and MrModelTest (phylogenetic analysis using parsimony (PAUP)) determined the best models was GTR+I+G; MEGA 6.06 (*Tamura et al., 2013*) was used to establish a phylogenetic tree based on the maximum likelihood (ML) (*Trifinopoulos et al., 2016*). The Bayesian inference (BI) tree was established using MrBayes (*Ronquist & Huelsenbeck, 2003*).

The divergence times of genus *Bottapotamon* were estimated from the combined 16S rRNA and mtDNA COI sequences, based on the Bayesian evolutionary analysis sampling trees (BEAST) program, and four calibration points were used. The Potamidae family has been divided into two major subfamilies, Potamiscinae and Potaminae, estimated to have a divergence time of 20.9–24.7 Ma, which was set as calibration point 1 in our

**Table 1 Specimens and GenBank accession numbers of genus *Bottapotamon*.**

| | Localities | Museum catalogue No. | Haplotypes | COI Accession No. | 16S Accession No. | H3 Accession No. |
|---|---|---|---|---|---|---|
| *Bottapotamon fukiense* | Shangshan Village, Zhenghe County, Fujian | NCU MCP4156 | Bfj1 | MK920086 | MK795653 | MK952581 |
| | Siqian Village, Shouning County, Fujian | NCU MCP4090 | Bfj2 | MK920087 | MK795654 | MK952582 |
| | Xiapu Village, Ningde County, Fujian | NCU MCP4089 | Bfj3 | MK920088 | MK795655 | MK952583 |
| | | NCU MCP4089 | Bfj4 | MK920089 | MK795656 | MK952584 |
| *Bottapotamon youxiense* | Xiwei Village, Youxi County, Fujian | NCU MCP4092 | Byx1 | MK920099 | MK795666 | MK952594 |
| | Xiwei Village, Youxi County, Fujian | NCU MCP4158 | Byx2 | MK920100 | MK795667 | MK952595 |
| | Xiwei Village, Youxi County, Fujian | NCU MCP4159 | Byx3 | MK920101 | MK795668 | MK952596 |
| *Bottapotamon engelhardti* | Chimu Village, Youxi County, Fujian | NCU MCP4091 | Bes1 | MK920081 | MK795648 | MK952576 |
| | Tangsan Village, Youxi County, Fujian | NCU MCP4157 | Bes2 | MK920082 | MK795649 | MK952577 |
| | | NCU MCP4157 | Bes3 | MK920083 | MK795650 | MK952578 |
| | | NCU MCP4157 | Bes4 | MK920084 | MK795651 | MK952579 |
| | | NCU MCP4157 | Bes5 | MK920085 | MK795652 | MK952580 |
| *Bottapotamon nanan* | iqian Village, Shouning County, Fujian | NCU MCP4090 | Bna1 | MK920093 | MK795660 | MK952588 |
| | | NCU MCP4090 | Bna2 | MK920094 | MK795661 | MK952589 |
| | Yongjia County, Zhejiang | NCU MCP4038 | Bna3 | MK920095 | MK795662 | MK952590 |
| | | NCU MCP4038 | Bna4 | MK920096 | MK795663 | MK952591 |
| | Yongjia County, Zhejiang | NCU MCP4039 | Bna5 | MK920097 | MK795664 | MK952592 |
| | | NCU MCP4039 | Bna6 | MK920098 | MK795665 | MK952593 |
| *Bottapotamon lingchuanense* | Bindong Village, Lingchuan County, Guangxi Zhuang Autonomous Region | NCU MCP3281 | Blc1 | MK920090 | MK795657 | MK952585 |
| | Yuanpu Village, Gongcheng County, Guangxi Zhuang Autonomous Region | NCU MCP4076 | Blc2 | MK920091 | MK795658 | MK952586 |
| | | NCU MCP4076 | Blc3 | MK920092 | MK795659 | MK952587 |
| *Bottapotamon chenzouense* sp.n. | Zixing County, Chenzhou City, Hunan | NCU MCP643 | Bcz1 | MK920079 | MK795646 | MK952574 |
| | | NCU MCP643 | Bcz2 | MK920080 | MK795647 | MK952575 |
| *Bottapotamon luxiense* sp.n. | Yixiantian Wugongshan Mountain, Luxi County, Pingxiang City, Jiangxi | NCU MCP4200 | Blx1 | MK993542 | MK981408 | MK993544 |
| | | NCU MCP4200 | Blx2 | MK993543 | MK981409 | MK993545 |

study (*Shih, Yeo & Ng, 2010*). From the Parathelphusidae subfamily, *Somanniathelphusa taiwanensis*, which is distributed in Taiwan Island and separated from *Somanniathelphusa amoyensis*, which is distributed in Fujian Province, for approximately 0.27–1.53 Ma (*Jia*
**Table 2  Primer sequences used in this study.**

| Gene | Primer name | Sequence (5′–3′) | Sequence length | Reference |
|---|---|---|---|---|
| COI | COI-1490 | GGTCAACAAATCATAAAGATATTGG | 750bp | *Folmer et al. (1994)* |
| | COI-2198 | TAAACTTCAGGGTGACCA AAAAATCA | | |
| 16S rRNA | 16S-1471 | CCTGTTTANCAAAAACAT | 550bp | *Crandall, Fitzpatrick & Faith (1996)* |
| | 16S-1472 | AGATAGAAACCAACCTGG | | |
| H3 | H3-F | ATGGCTCGTACCAAGCAGACVGC | 374bp | *Colgan et al., 1998* |
| | H3-R | ATATCCTTRGGCATRATRGTGAC | | |

*et al., 2018*). This is consistent with the quaternary glacial period and interglacial period and agrees with the separation of Taiwan Island and Fujian Province; this time point was set as calibration point 2. In the geological area where genus *Bottapotamon* is distributed, the Wuyi Mountains gradually formed by the compression of the Pacific plate and the Indian plate in the Neogene-Quaternary (1.64–23.3 Ma) (*Li, 1984*); this time point was set as calibration point 3. A Yule speciation model was constructed for speciation within the genus *Bottapotamon*. We used a GTR+G model with parameters obtained from MrMTGui: ModelTest and MrModelTest (PAUP) for each gene. Seventeen independent MCMC chains were run for 200,000,000 generations, and every 20,000 generations were sampled. The convergence of the 17 combined chains was determined by the evolutionary stable strategy (ESS) (>200 as recommended) for each parameter in Tracer after the appropriate burn-in and cutoff (default of 10% of sampled trees). Trees in the 17 chains were combined using LogCombiner (v.1.6.1, distributed as part of the BEAST package) and were assessed using TreeAnnotator (v.1.6.1, distributed as part of the BEAST package). A chronogram was constructed by FigTree.

## Nomenclatural note

The electronic version of this article in Portable Document Format (PDF) will represent a published work according to the International Commission on Zoological Nomenclature (ICZN), and hence the new names contained in the electronic version are effectively published under that Code from the electronic edition alone. This published work and the nomenclatural acts it contains have been registered in ZooBank, the online registration system for the ICZN. The ZooBank LSIDs (Life Science Identifiers) can be resolved and the associated information viewed through any standard web browser by appending the LSID to the prefix http://zoobank.org/. The LSID for this publication is: [urn: lsid: zoobank.org: pub:211926FF-6950-4DFE-95C4-F5247CA9E0BA]. The online version of this work is archived and available from the following digital repositories: Peer J, PubMed Central and CLOCKSS.

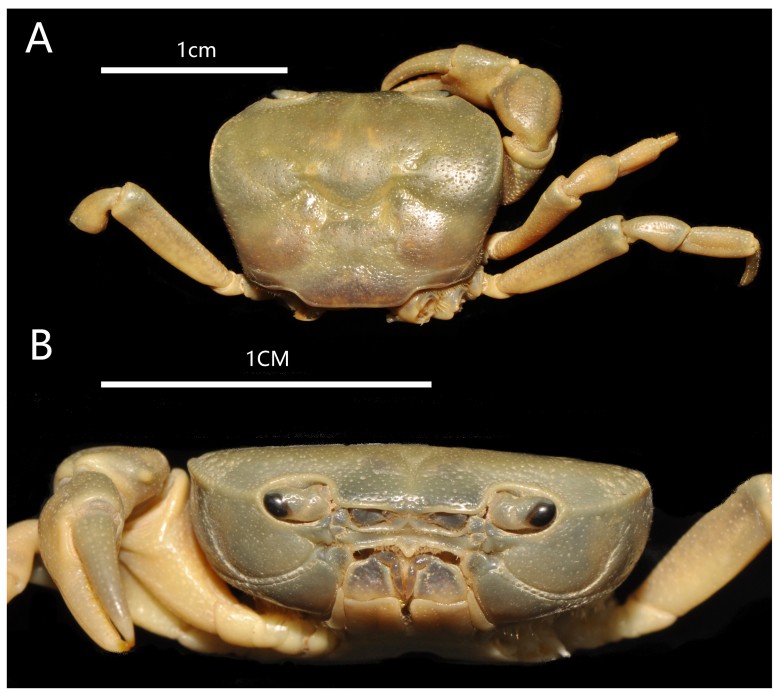

**Figure 2** ***Bottapotamon chenzhouense*** **sp. n. Holotype male (20.67 × 15.60 mm) (NCU MCP 643).** (A) Overall habitus; (B) frontal view of cephalothorax. Photograph taken by Jie-Xin Zou, November 2018.

## RESULTS

### Systematics

Potamidae Ortmann, 1896
*Bottapotamon* Tüerkay & Dai, 1997

***Bottapotamon chenzhouense*** **sp. n. Gao, Cui & Zou (Figs. 2–6)**
urn: lsid zoobank. org: art: E43C4BBB-E429-4C17-8ACD-E4295F426BCB

### Materials examined

Holotype: 1♂ (20.67 × 15.60 mm) (NCU MCP 643), Huangcao Village, Chenzhou City, Hunan Province, China, 25°39′24.60″N, 113°30′4.07″E, 141 m asl. Coll. Ding-mei Luo, July 26th, 2006. Paratypes: 1 ♀ (18.64 × 14.62 mm) (NCU MCP 643), the same data as the holotype.

### Comparative materials

*B. fukiense* (*Dai et al., 1979*): 2♂ (15. 66 × 12.64 mm, 13.15 × 10.26 mm) (NCU MCP 4089), Xiapu Village, Ningde County, Fujian Province; 1♂ (13. 26 × 11.05 mm) (NCU MCP 4156), Shangshan Village, Zhenghe County, Fujian Province; 1♂ (22.93 × 17.67 mm) (NCU MCP 4090), Siqian Village, Shouning County, Fujian Province; 1 ♀ (19.26 × 15.70 mm) (NCU MCP 4090), Shangshan Village, Zhenghe County, Fujian Province. *B. engelhardti* (*Bott, 1967*): 3♂♂ (15.32 × 11.90 mm, 17.08 × 13.46 mm, 18.85 × 15.01 mm) (NCU

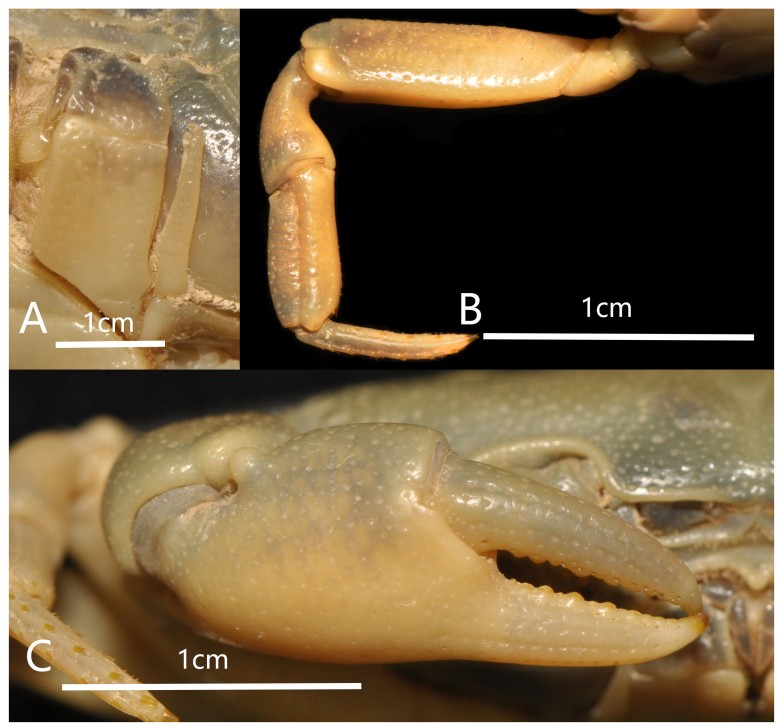

**Figure 3** ***Bottapotamon chenzhouense* sp. n. Holotype male (20.67 × 15.60 mm) (NCU MCP 643).** (A) Left third maxilliped; (B) right fourth ambulatory leg; (C) outer view of chelipeds. Photograph taken by Jie-Xin Zou, November 2018.

MCP 4157), Tangsan Village, Youxi County, Fujian; 3♂♂ (16.23 × 13.78 mm, 17. 50 × 14.41 mm, 14.86 × 11.18 mm) (NCU MCP 4091), Chimu Village, Youxi County, Fujian Province; 1♀ (28.03 × 21.97 mm) (NCU MCP 4091), Chimu Village, Youxi County, Fujian Province. *B. yonganense* (*Cheng, Lin & Luo, 1993*): 1♂ (22.97 × 18.19 mm) (NCU MCP 4096), Sanming City, Fujian; *B. lingchuanense* (*Türkay & Dai, 1997*), 6♂♂ (24.36 × 19.51 mm, 22. 34 × 18.70 mm, 23.03 × 18.51 mm, 25.33 × 19.46 mm, 24.92 × 19.10 mm, 18.04 × 14.41 mm) (NCU MCP 4076), Yuanpu Village, Gongcheng County, Guangxi Zhuang Autonomous Region; 4♂♂ (19.36 × 15.55, 19.56 × 15.69 mm, 19.68 × 16.15 mm, 20.11 × 15.98 mm) (NCU MCP 3281), Bindong Village, Lingchuan County, Guangxi Zhuang Autonomous Region; 3♀♀ (20.94 × 16.27 mm, 19.87 × 16.29 mm, 22.19 × 17.73 mm, 20.22 × 15.97 mm), (NCU MCP 3281), Bindong Village, Lingchuan County, Guangxi Zhuang Autonomous Region. *B. youxiense* (*Cheng, Lin & Li, 2010*): 4♂♂ (14.27 × 12.21 mm, 13.57 × 11.05 mm, 13.78 × 11.16 mm, 14.09 × 11.42 mm) (NCU MCP 4092), 2♂ (13.35 × 10.60 mm, 13.41 × 11.02 mm) (NCU MCP 4158) . *B. nanan* (*Zhou, Zhu & Naruse, 2008*): 2♂ (28.48 × 22.65 mm, 22.23 × 16.92 mm) (NCU MCP 4090), Siqian Village, Shouning County, Fujian Province; 3♂♂ (23.59 × 18.92 mm, 21.73 × 17.36 mm, 22. 98 × 17.38 mm) (NCU MCP 4038), Yongjia County, Zhejiang Province; 2♂ (17.49 × 13.60 mm, 21. 28 × 16.11 mm), Yongjia County, Zhejiang Province; 1 ♀ (20.01 × 15.01 mm) (NCU MCP 4039), Yongjia County, Zhejiang Province.

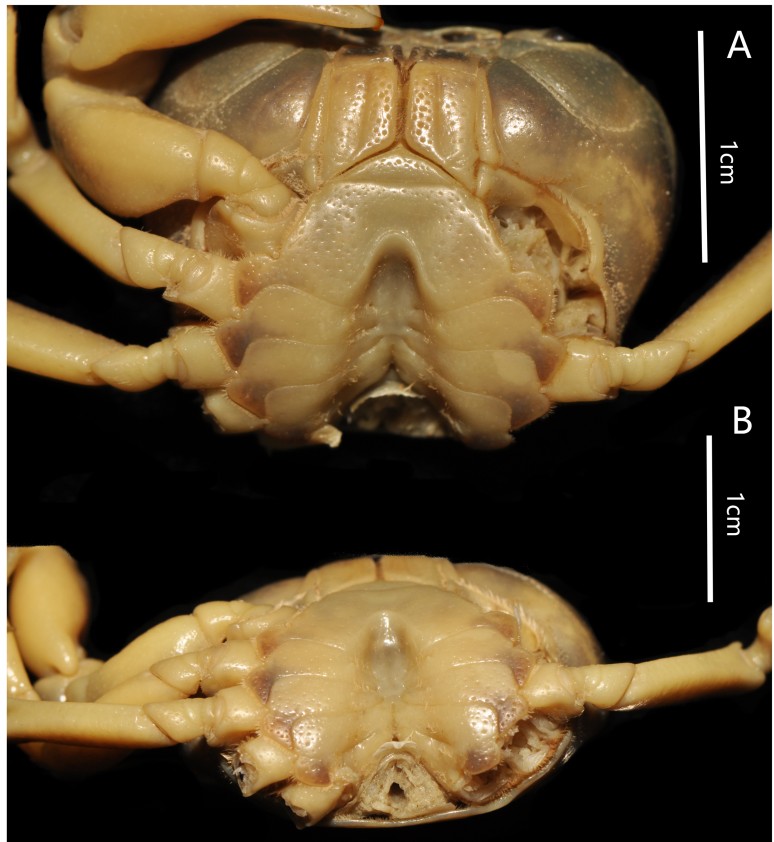

**Figure 4** ***Bottapotamon chenzhouense* sp. n. Holotype male (20.67 × 15.60 mm) (NCU MCP 643).** (A) Male sternum. Interruption between sutures of sternites 4/5, 5/6, 6/7; tubercle of abdominal lock. (B) Median logitudinal suture of sternites7, 8. Photograph taken by Jie-Xin Zou, November 2018.

## Diagnosis

Carapace subquadrate, flat, dorsal surface smooth (Fig. 2); approximately about 1.3 times broader than long; third maxilliped ischium about 1.5 times as long as broad, exopod without flagellum (Fig. 3A); male pleon triangular, sixth somite width 2.5 times length; telson triangular, tip rounded, with proximal width 1.7 times length; median groove of male thoracic sternum deep, interruption between sutures of sternites 4/5, 5/6, 6/7 broad (Fig. 4). G1 long, tip of terminal segment reaching beyond suture between thoracic sternites 4/5 *in situ*; subterminal segment 1.3 times as long as terminal segment; terminal segment slightly elongated, curved inward, distal part of terminal segment elongated with anterioventrally directed semicircular lobe. Female vulvae partially exposed anteriorly to the thoracic sternites 5/6 *in situ*, ovate, deep, posteromesial margin with a low raised rim, opened inward.

## Description

Carapace approximately about 1.3 times broader than long, dorsal surface gently convex from frontal view, regions not prominently inflated; with surface slightly pitted. Cervical

groove shallow, indistinct. H-shaped groove between the gastric region and cardiac region shallow but distinct. Postfrontal lobe blunt, separated medially by a Y-shaped groove extending to frontal region; postorbital crest indistinct, postorbital region slight concave. Frontal region deflexed downwards. Dorsal orbital margin ridged, external orbital angle triangular outer margin smooth; Anterolateral margin cristate, epibranchial tooth pointed, indistinct, clearly demarcated from external orbital tooth (Fig. 2).

Third maxilliped merus about 1.3 times as broad as long; Ischium about 1.5 times as long as broad, with distinct median sulcus; exopod reaching proximal third of merus length, without flagellum (Fig. 3A).

Male sternum pitted, sternites 1, 2 fused to form triangular structure; sternites 2, 3 separated by continuous suture; boundary between sternites,3, 4 present. Male sterno-pleonal cavity broad, shallow, with narrow median interruption in sutures 4/5, 5/6, 6/7; median line between sternites 7, 8 moderately short; male pleonal locking tubercle on posterior third of sternite 5 (Fig. 4).

Cheliped slightly unequal; margins crenulated; carpus with sharp spine on inner distal angle, with spinule at base; outer surface of manus with convex granules, manus about 1.6 times as long as high, slightly longer than movable finger, gape wide when fingers closed, cutting edge lined with low teeth (Fig. 3C).

Ambulatory legs slender; margins of propodus smooth; last leg with propodus about 1.8 times as long as broad, slightly shorter than dactylus (Fig. 3B).

G1 slender, ventral flap with transparent protrusion, with a fold covering the surface of theentire subterminal. Tip of terminal segment slightly reaching beyond sternal pleonal locking structure *in situ*, subterminal segment about 1.3 times as long as terminal segment. G1 slightly curved anterioventrally; distal part of G1 terminal segment distinctly broader than proximal part. G2 subterminal segment about 2.3 times as long as terminal segment (Figs. 5A and 6A).

## Remarks

The new species fits well within the morphological definition of the genus *Bottapotamon* (*Türkay & Dai, 1997*; *Cheng, Lin & Li, 2010*; *Zhou, Zhu & Naruse, 2008*): G1 is slender, tip of terminal segment reaching suture between thoracic sternites 4/5 *in situ*; terminal segment slightly elongated inward (Table 3). Nonetheless, the new species can be distinguished from co - genus, by the carapace surface gently convex, cervical groove indistinct; H-shaped groove shallow but distinct; epibranchial tooth pointed and indistinct, third maxilliped without flagellum; chelipeds carpus with sharp spine on inner distal angle; and the ventromedially curved G1, which subterminal segment about 1.3 times as long as terminal segment (Table 3). The most obvious specific character of the new species is that the ventral flap of G1 with transparent protrusion, with a fold covering the surface of the entire subterminal region (Figs. 5A and 6A).

## Etymology

The species is named after the type locality: Chenzhou city, Hunan Province, China.

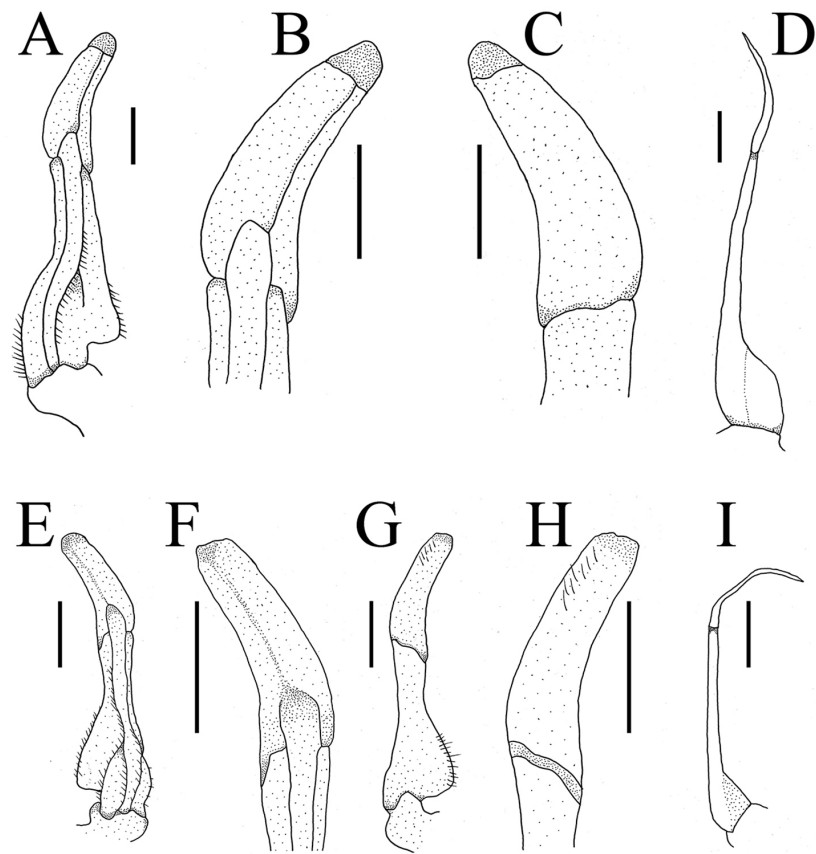

**Figure 5** **Gonopods.** (A–D) *Bottapotamon chenzhouense* sp .n. Holotype male (20.67 × 15.60 mm) (NCU MCP 643); (E–I) *Bottapotamon luxiense* sp. n. Holotype male (17.36 × 13.26 mm) (NCU MCP 4200).

## Distribution

*B. chenzhouense* sp. n. was found under stones in a mountain stream in Huangcao village, Sunxian District, Chenzhou City, Hunan Province, China.

### *Bottapotamon luxiense* sp. n. Gao, Cui & Zou (Figs. 5–10)

urn: lsid zoobank. org: art: 1C1CC520-193A-405E-9A2D-DC79E7D4AA87.

### Materials examined

Holotype: 1♂ (17. 36 × 13.26 mm) (NCU MCP 4200), Yixiantian Wugongshan Mountain, Luxi County, Pingxiang City, Jiangxi Province, China, 27° 28′56.16″N, 114°10′27.51″E, 1331 m asl. Coll. Song-bo Wang, May 6th, 2019. Paratypes: 1♂ (19. 21× 14.67 mm) (NCU MCP 4200). Others: 10 ♀♀ (17. 51× 13.89 mm, 14. 43× 11.30 mm, 17. 93× 14.23 mm, 18. 08× 14.39 mm, 19. 61× 15.58 mm, 16. 77× 12.74 mm, 15. 88× 12.00 mm, 17. 40× 13.77 mm, 16. 36× 12.93 mm, 19. 09× 15.02 mm) (NCU MCP 4200), 14♂♂ (17. 33×13.76 mm, 16. 10× 12.93 mm, 14. 61×12.10 mm, 15. 03× 11.27 mm, 12. 01× 9.24 mm, 12. 01×9.48 mm, 10. 59×8.33 mm, 12. 61× 10.39 mm, 13. 53× 10.89 mm, 14. 12× 11.24

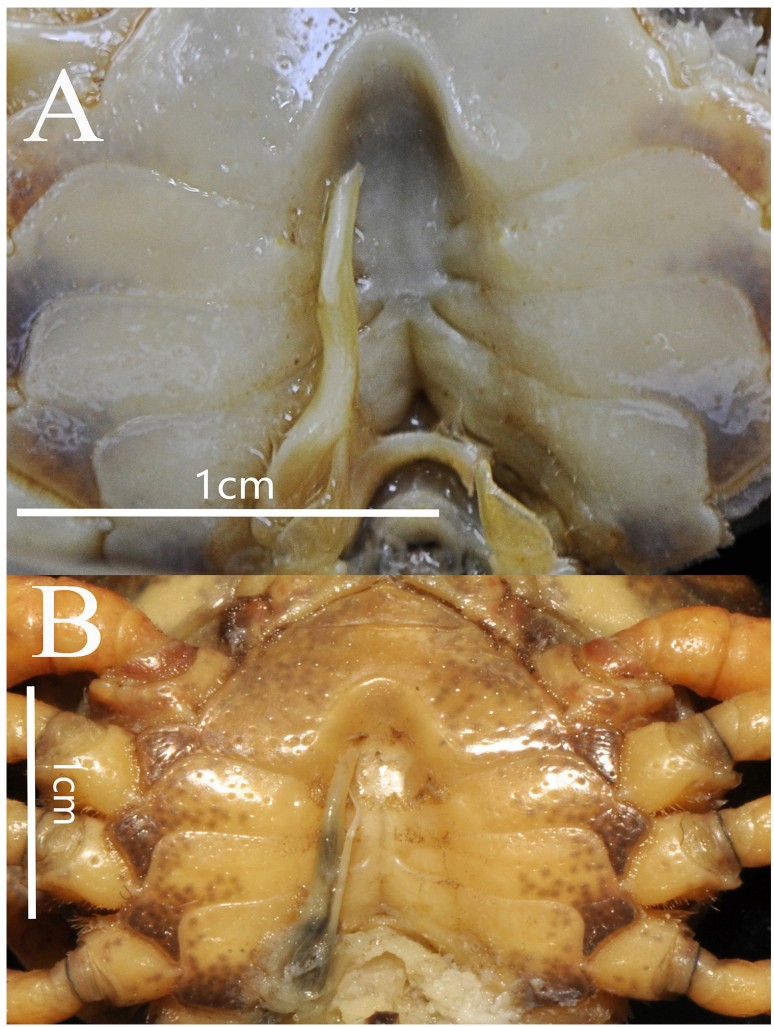

**Figure 6** **Natural position of male G1 and median longitudinal suture of sternites 7,8.** (A) *Bottapotamon chenzhouense* sp. n. Holotype male (20.67 × 15.60 mm) (NCU MCP 643); (B) *Bottapotamon luxiense* sp. n. Holotype male (17.36 × 13.26 mm) (NCU MCP 4200). Photograph taken by Jie-Xin Zou, November 2018.

mm, 12. 84×10.07 mm, 12. 15× 9.76 mm, 14. 31× 11.64 mm, 11. 71× 9.20 mm) (NCU MCP 4200), the same data as holotype.

## Comparative materials

Same as *Bottapotamon chenzhouense* sp. n.

## Diagnosis

Carapace about 1.3 times broader than long, subquadrate, flat, dorsal surface gently convex longitudinally; cervical groove distinct, H-shaped groove between gastric, cardiac regions distinct (Fig. 7); third maxilliped ischium about 1.5 times as long as broad, with flagellum (Fig. 8A); male abdomen broadly triangular, telson triangular , with about 1.6 times as broad as long (Fig. 6B); median groove of male thoracic sternum deep, interruption

Gao et al. (2019), *PeerJ*, DOI 10.7717/peerj.7980

**Table 3   Primer sequences used in this study.**

| Species | *B. fukiense* | *B. yonganense* | *B. engelhardti* | *B. nanan* | *B. youxiense* | *B. lingchuanense* | *B. chenzhouense* sp. n | *B. luxiense* sp. n |
|---|---|---|---|---|---|---|---|---|
| Carapace | Flat, cervical groove indistinct | Swollen, cervical groove distinct | Swollen, cervical groove indistinct | Swollen, cervical groove distinct. | Swollen, cervical groove indistinct | Swollen, cervical groove indistinct | Swollen, cervical groove indistinct | Swollen, cervical groove distinct. |
| External orbital angle | Blunt | Triangle | Blunt | Blunt | Triangle | Triangle | Triangle | Triangle |
| Third maxilliped merus | Length to width ratio 1.3 | Length to width ratio 1.1 | Length to width ratio 1.2 | Length to width ratio 1.4 | Length to width ratio 1.1 | Length to width ratio 1.2 | Length to width ratio 1.3 | Length to width ratio 1.4 |
| Male abdomen | Broad triangular | Narrow triangular | Broad triangular | Broad triangular | Broad triangular | Broad triangular | Narrow Triangular | Broad triangular |
| Male abdomen telson | Width to length ratio 1.5 | Width to length ratio 1.3 | Width to length ratio 1.3 | Width to length ratio 1.4 | Width to length ratio 1.5 | Width to length ratio 1.2 | Width to length ratio 1.3 | Width to length ratio 1.3 |
| Immovable finger | Length to width ratio 1.3 | Length to width ratio 1.7 | Length to width ratio 1.4 | Length to width ratio 1.7 | Length to width ratio 1.7 | Length to width ratio 1.4 | Length to width ratio 1.4 | Length to width ratio 1.8 |
| G1 | Stout, straight | Slender, distal segment tabular arcuate | Slender, distal dorsal lobe convex | Slender, distinct longitudinal groove | Slender, distal segment spacious and strong | Slender, terminal, segment tortuous slightly | Slender, ventral flap with transparent protrusion | Blunt |

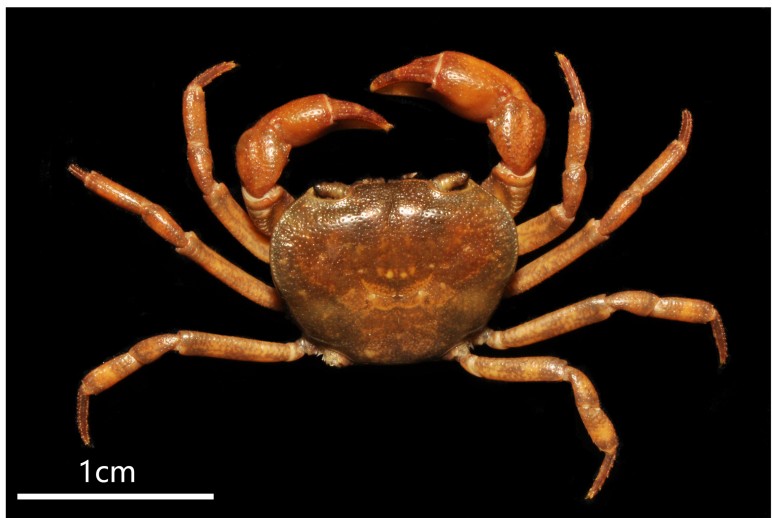

**Figure 7 *Bottapotamon luxiense* sp. n. Holotype male (17.36 × 13.26 mm) (NCU MCP 4200-Blx1).** Overall habitus. Photograph taken by Jie-Xin Zou, May 2019.

between sutures of sternites 4/5, 5/6, 6/7 broad. G1 long and blunt, tip of terminal segment reaching suture between thoracic sternites 4/5 *in situ*; subterminal segment 1.2 times as long as terminal segment; terminal segment slightly elongated inward, distal part of terminal segment elongated with anterioventrally directed semicircular lobe. Female vulvae partially exposed anteriorly to the thoracic sternites 5/6 *in situ*, ovate, deep, posteromesial margin with a low raised rim, opened inward.

## Description

Carapace nearly ellipse in shape, about 1.3 times broader than long, flat, dorsal surface punctate, glabrous; regions distinctly defined; epibranchial region rugose, mesogastric regionslightly convex. Cervical groove distinct. H-shaped groove between the gastric region and cardiac region shallow but distinct. Postfrontal lobe blunt; postorbital crest indistinct, postorbital region slight concave. Frontal region deflexed downwards. Dorsal orbital margin ridge, external orbital angle triangular, outer margin smooth. Anterolateral margin cristate, epibranchial tooth pointed (Fig. 7).

Third maxilliped merus trapezoidal about 1.4 times as broad as long; ischium about 1.5 times as long as broad, with distinct median sulcus; exopod reaching proximal third of merus length, with flagellum (Fig. 8A).

Thoracic sternum pitted; sternites 1/2 completely fused to form triangular structure; sternites 2/3 separated by continuous suture; boundary between sternites 3/4 present, indistinct. Sterno-pleonal cavity broad, shallow, with narrow median interruption in sutures 4/5, 5/6, 6/7; median line between sternites 7/8 moderately long (Fig. 9).

The male sternum is relatively flat with numerous small pits; sternites 1/2 fused triangular; transverse sulcus between sternites 2/3 suture; sternites 3/4 fused without obvious demarcation. Male sterno-pleonal cavity is medium in depth wide; median

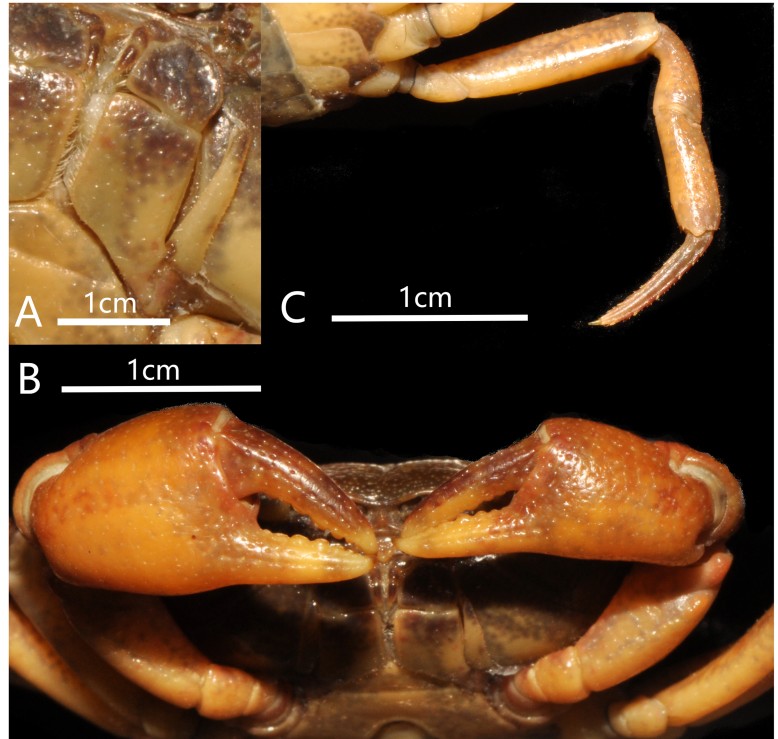

**Figure 8** *Bottapotamon luxiense* **sp. n. Holotype male (17.36 × 13.26 mm) (NCU MCP 4200).** (A) Left third maxilliped; (B) outer view of chelipeds; (C) right fourth ambulatory leg. Photograph taken by Jie-Xin Zou, May 2019.

longitudinal groove between sternites 7/8 short; male pleonal locking tubercle on posterior third of sternite 5 (Fig. 6B).

Chelipeds slightly unequal; outer surface of manus with granules, manus about 1.5 times as long as high, slightly longer than movable finger, gape wide when fingers closed, cutting edge lined with low teeth (Fig. 8B).

Ambulatory legs slender; margins of propodus smooth; last leg with propodus about 1.7 times as long as broad, slightly shorter than dactylus (Fig. 8C).

G1 blunt, tip of terminal segment slightly reaching beyond sternal pleonal locking structure *in situ*, subterminal segment about 1.4 times as long as terminal segment. G1 slightly curved ventrolaterally; distal part of G1 terminal segment distinctly broader than proximal part. G2 subterminal segment about 2.2 times as long as terminal segment (Figs. 5B and 6B).

## Remarks

The new species fits well within the morphological definition of the genus *Bottapotamon* (*Türkay & Dai, 1997*; *Cheng, Lin & Li, 2010*; *Zhou, Zhu & Naruse, 2008*), especially similar to *B. fukiense*, and *B. lingchuanense* in shape of carapace and slender G1. With regards to the other species of genus *Bottapotamon*, they can be separated (Table 3). Adult male specimens of *B. luxiense* sp. n. have the gastric regions relatively smooth with the rest of the

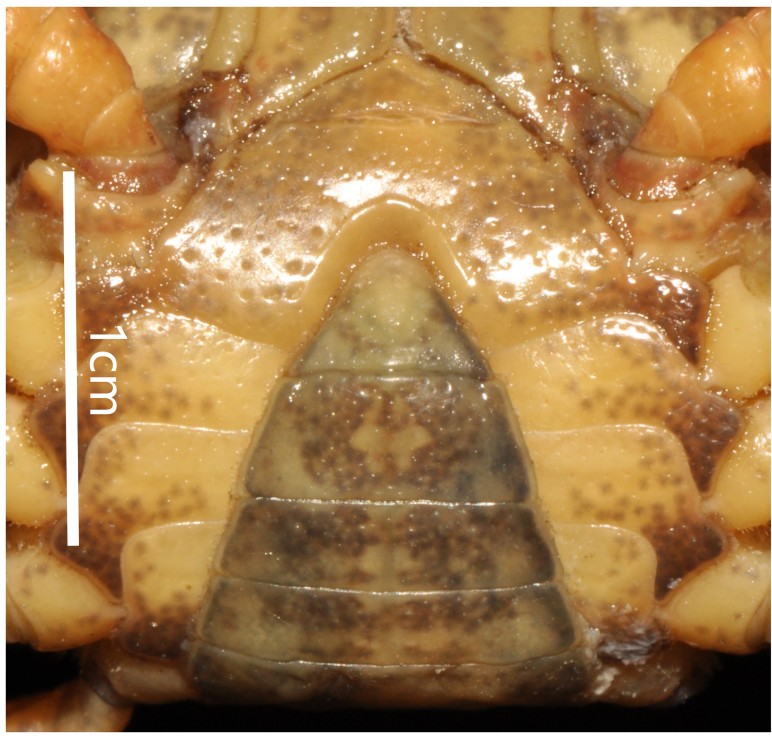

**Figure 9** ***Bottapotamon luxiense*** **sp. n. Holotype male (17.36 × 13.26 mm) (NCU MCP 4200-Blx1).** Male sternum. Photograph taken by Jie-Xin Zou, May 2019.

surfaces also some rugose and granulose; H-shaped groove shallow but distinct (Fig. 7). The G1 of *B. luxiense* sp. n. is also quite dfferent with the terminal segment straight, slender and blunting towards the tip (Figs. 5B and 6B); third maxilliped with flagellum; median longitudinal groove between sternites 7/8 short; chelipeds carpus with sharp spine on inner distal angle, with spines at base (Fig. 8B).

## Etymology
The species is named after the type locality: Yixiantian Wugongshan Mountain, Luxi County, Pingxiang City, Jiangxi Province, China.

## Living coloration
The dorsal surfaces of the carapace and pereopods are dark purple-red, and the joints of the cheliped merus and carpus the ambulatory legs are bright red. The inner surface of the immovable finger and distal part of the movable finger are almost milky.

## Distribution
*B.luxiense* sp. n. was found under stones in a mountain stream in Yixiantian Wugongshan Mountain, Luxi County, Pingxiang City, Jiangxi Province, China (Fig. 10).

## Ecology
*B. chenzhouense* sp. n. and *B. luxiense* sp. n. were collected in the Luoxiao mountains. This region has a humid subtropical monsoon climate and is in the Xiangjiang River and

Ganjiang River watershed, which has rich biodiversity (*Wang, 1998*). Similar to the natural habitat of other *Bottapotamon* species, *B. chenzhouense* sp. n. and *B. luxiense* sp. n. can be found under small rocks in sandy creek beds in narrow mountain streams or highway drains with clear, slow flowing and cool water surrounded by dwarf shrubs or grasses (Fig. 10).

**Phylogenetic analyses and Divergence time estimation**

Within genus *Bottapotamon*, a 1323 bp segment (excluding the primer regions) of the combined mtDNA COI, 16S rRNA and nuclear histone H3 from all 25 specimens was analysed. The phylogenetic trees were constructed by ML analysis, and the corresponding support values were calculated by ML and BI analyses, both of which had high support values. The results showed that the genus *Bottapotamon* is monophyletic, and confirmed that *B. chenzhouense* sp. n. and *B. luxiense* sp. n. are new species of genus *Bottapotamon* and supported the relationship of the genus *Bottapotamon* (Fig. 11). With regard to the relationships among the all specimens, the phylogenetic tree also show some distinct geographical distibution (Fig. 1). *B. engelhardti*, *B. yonganense* and *B. nanan*, which are mostly distributed in the Wuyi Mountain Range, form a clade; *B. luxiense* sp. n. forms a sister clade to the clade of *B. engelhardti*, *B. yonganense* and *B. nanan*. The next sister clade is composed of *B. chenzhouense* sp. n., which is distributed in the Luoxiao Mountain Range, and the furthest sister clade is composed of *B. lingchuanense,* which is situated some diatance from the Wuyi Mountain Rnage and Luoxiao Mountain Range, but near the Nanling Mountain. However, *B. fukiense* and *B. youxiense* are also distributed in the Wuyi Mountain Range, they do not assemble with *B. engelhardti*, *B. yonganense* and *B. nanan*.

Based on the relaxed molecular clock estimation, the earliest divergence time for genus *Bottapotamon was* estimated to be 3.49–1.08 Ma. The divergence time estimation results are consistent with the four calibration points. *B. fukiense* and *B. youxiense* diverged 1.96 Ma (95% confidence interval = 2.65–1.31 Ma), *B. luxiense* diverged 1.90 Ma (95% confidence interval = 2.05–1.09 Ma), *B. lingchuanense* and *B. chenzhouense* sp. n. diverged 1.51 Ma (95% confidence interval = 1.6–0.7 Ma); *B. engelhardti* and *B. nanan* diverged 1.08 Ma (95% confidence interval = 1.76–0.80 Ma) (Fig. 12).

## DISCUSSION

In mainland China, the genus *Bottapotamon* is primarily distributed in the Wuyi Mountain Range area; *B. luxiense* sp. n., *B. youxiense, B. nanan, B. engelhardti* and *B. yonganense* are restricted within an area east of the Wuyi Mountain Range (Fig. 1). There is no record of any of these five species in Jiangxi, despite extensive surveys of this area by the authors and their colleagues over many years (*Dai, 1999*; *Shi et al., 2012*). The altitude of the Wuyi Mountain Range is clearly high enough to prevent these species from reaching Jiangxi. *B. fukiense* occurs on both sides of the Wuyi Mountain Range (Fujian and Jiangxi Provinces), and is able to disperse across these mountains. The divergence time of *B. fukiense* is 1.96 Ma (95% confidence interval = 2.65–1.31 Ma) (Fig. 12), and the divergence time agrees well with records of the Pacific plate and Indian plate extrusion in the Neogene-Quaternary (1.64–23.3 Ma) (*Li, 1984*). Therefore, these geological events may explain the distribution

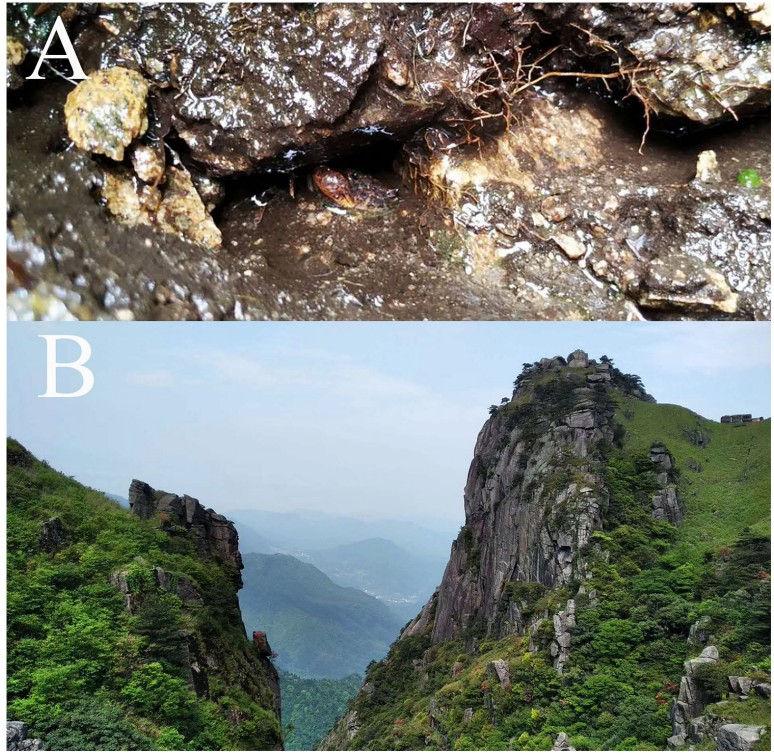

**Figure 10** **The type locality of *Bottapotamon luxiense* sp. n.** (A) Living under rocks. (B) Surroundings of type locality. Photo taken by Song-bo Wang, May 2019.

pattern of the genus *Bottapotamon* in the Wuyi Mountain Range. The ancestor of *B. fukiense* originated in an area close to the Wuyi Mountains, which probably dispersed across the Wuyi Mountain Range when it was still a lowland, before the Wuyi Mountain Formation and smaller-scale mountain deformations occured and separated.

In the Nanling mountain range, unique karst formation and the south Asian subtropical humid monsoon climate conditions provide a good living environment for all types of wildlife, including freshwater crabs. However, only one species of the genus *Bottapotamon*, *B. lingchuanense*, was isolated in this area, and there is an 830 km gap between *B. lingchuanense* and other species distributed within the Wuyi Mountain Range (Fig. 1), which has always been the focus of researches on the genus *Bottapotamon*. This study reports two new species of genus *Bottapotamon*, *B. chenzhouense* sp. n., which was first discovered in Chenzhou City, Hunan Province, in south of Luoxiao Mountains, and *B.luxiense* sp. n., which is distributed in north of the Luoxiao Mountains (Fig. 1). Divergence time estimation results suggested that *B. chenzhouense* sp. n., *B. luxiense* sp. n., and *B. lingchuanense* were isolated at almost the same time (*B. luxiense* sp. n. diverged 1.90 Ma, and *B. lingchuanense* and *B. chenzhouense* sp. n. diverged at 1.51 Ma) (Fig. 12). The authors speculated that the Luoxiao Mountains continuously rose due to neotectonic movement and gradually formed the Xiangjiang River and Ganjiang River watershed (*Wang, 1998*). The ancestors of the genus *Bottapotamon* occurred on both sides of the Luoxiao Mountains during the

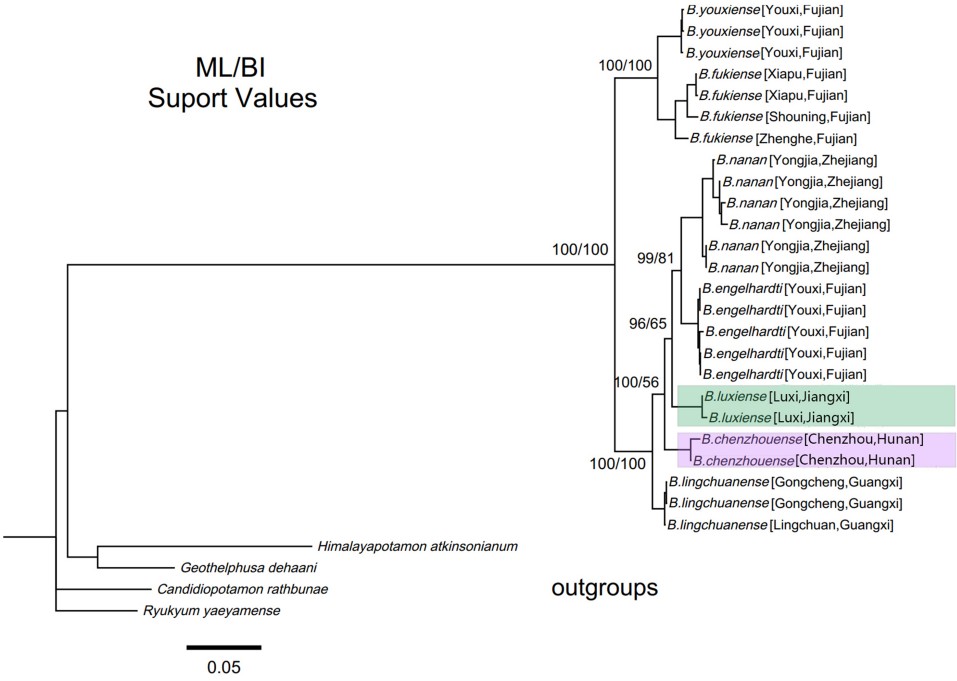

**Figure 11 Phylogenetic tree of the genus *Bottapotamon*.** A maximum likelihood (ML) tree of the genus *Bottapotamon*, and outgroups, based on the combined mtDNA COI, 16S rRNA and nuclear histone H3 genes (length = 1,404 bp). Support values ($P \geqq 50\%$) for ML, BI is represented at the nodes. Locality names in Table 1 are parenthesized behind specimens.

mountains formation process, and under the influence of karst landforms and the Danxia landform, gradually isolated *B. luxiense* sp. n., *B. chenzhouense* sp. n. and *B. lingchuanense*. In addition, the climatic conditions in this area are ideal for *Bottapotamon*. The authors speculate that many new species of the genus *Bottapotamon* are likely to exist in the region from the Wuyi Mountain Ranges to the Nanling Mountain Range, but get to be discovered.

## CONCLUSIONS

*Bottapotamon chenzhouense* sp. n. and *B. luxiense* sp. n., two new species from the Luoxiao Mountains were reported in this paper. These two new species compensated for the geographical gap in the genus *Bottapotamon*, and confirm the independence and intra- and interspecific relationships of genus *Bottapotamon*. Combined with estimates of divergence times, this paper suggests that the genus *Bottapotam* was formed at 3.49–1.08 Ma. Molecular evidence further supports the scientific hypothesis of the authors that genus *Bottapotamon* originated on both sides of the Wuyi Mountains and Luoxiao Mountains. In the geological area where the genus *Bottapotamon* is distributed, the Wuyi Mountains gradually formed offshore and inland of southeastern China by the compression of the Pacific plate and the Indian plate in the Neogene-Quaternary, and the Luoxiao Mountains formed continuously in the north-south direction because of neotectonic movement. Thus, the geographical

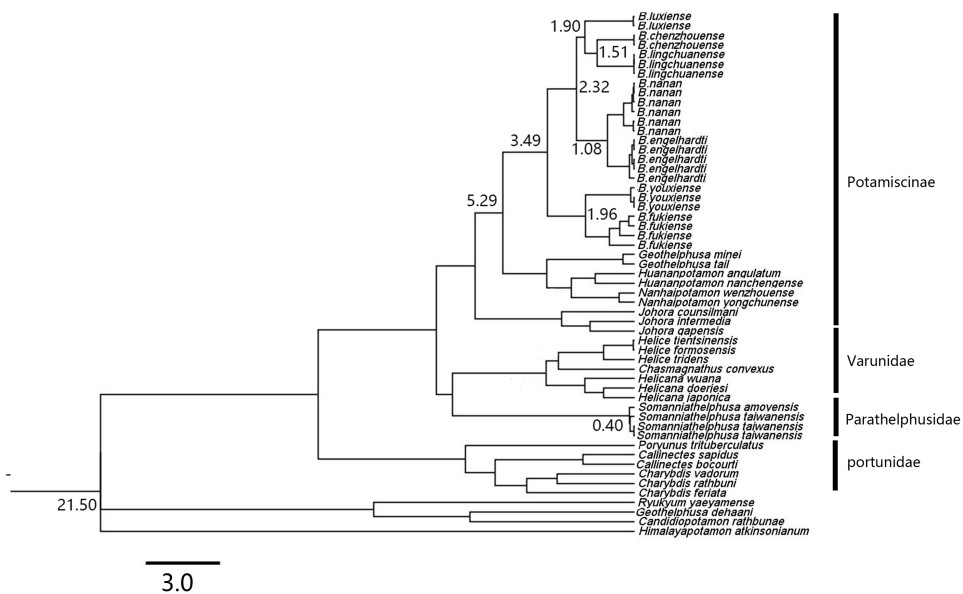

**Figure 12 A chronogram of the genus *Bottapotamon*.** Based on the mtDNA COI, 16S rRNA genes. The divergence times for genus *Bottapotamon* and calibration points are shown at the main nodes. Calibration point 1 was set for the divergence time between subfamily Potamiscinae and subfamily Potaminae (estimated value = 21.50 Ma); calibration point 2 was set for the glacial periods in Taiwan Strait (*Somanniathelphusa taiwanensis* and *Somanniathelphusa amoyensis*, estimated value = 0.40 Ma); Formation time of Wuyi mountains was set for calibration point 3 (The divergence time of *B. fukiense* is 1.96 Ma).

distribution patterns of the genus *Bottapotamon* was formed gradually with the various events.

# ACKNOWLEDGEMENTS

We thank Mao-rong Cai, Yi-yang Xu, Yu-Jie Zhao and Hua Guo for collecting the specimens of the new species. Special thanks are expressed to Xin-nan Jia and Shu-xin Xu for their help and advice on the manuscript. We would also like to thank Professor Xian-min Zhou for his guidance in this study.

## Funding

The work is supported by the National Sharing Service Platform for Parasite Resources (TDRC-22), the National Natural Science Foundation of China (No. 31560179, 31460156), the Natural Science Foundation of Jiangxi Province (No. 20171BAB205108) and the Student's Platform for Innovation and Entrepreneurship Training Program (No. 2017262). The funders had no role in study design, data collection and analysis, decision to publish, or preparation of the manuscript.

## Grant Disclosures

The following grant information was disclosed by the authors:

National Sharing Service Platform for Parasite Resources: TDRC-22.

National Natural Science Foundation of China: 31560179, 31460156.

Natural Science Foundation of Jiangxi Province: 20171BAB205108.

Student's Platform for Innovation and Entrepreneurship Training Program: 2017262.

## Competing Interests

The authors declare there are no competing interests.

## Author Contributions

- Ning Gao conceived and designed the experiments, performed the experiments, analyzed the data, prepared figures and/or tables, authored or reviewed drafts of the paper, approved the final draft.
- Ying-Yi Cui conceived and designed the experiments, performed the experiments, authored or reviewed drafts of the paper, approved the final draft.
- Song-Bo Wang conceived and designed the experiments, contributed reagents/materials/analysis tools, prepared figures and/or tables, authored or reviewed drafts of the paper, approved the final draft.
- Jie-Xin Zou conceived and designed the experiments, analyzed the data, contributed reagents/materials/analysis tools, prepared figures and/or tables, authored or reviewed drafts of the paper, approved the final draft.

## DNA Deposition

The following information was supplied regarding the deposition of DNA sequences:

The group genus Bottapotamon sequences are available at GenBank accession numbers MK795646 to MK795668 , MK920079 to MK920101 , MK952547 to MK952596 , MK981408 to MK981409 , MK993542 to MK993545.

## Data Availability

The raw data is available at Genbank: MK795646–MK993545.

All specimens in this study are housed in the permanent collections at the Department of Parasitology, Medical College of Nanchang University (NCUMCP). Specimen numbers can be found in Table 1.

## New Species Registration

The following information was supplied regarding the registration of a newly described species:

Publication LSID: urn:lsid:zoobank.org:pub:211926FF-6950-4DFE-95C4-F5247CA9E0BA

Bottapotamon chenzhouense sp. n. LSID: urn:lsid:zoobank.org:art:E43C4BBB-E429-4C17-8ACD-E4295F426BCB.

Bottapotamon luxiense sp. n. LSID: urn:lsid:zoobank.org:art:1C1CC520-193A-405E-9A2D-DC79E7D4AA87.

## Supplemental Information

Supplemental information for this article can be found online at http://dx.doi.org/10.7717/peerj.7980#supplemental-information.

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
