# Peer review of "Two new species and the molecular phylogeography of the freshwater crab genus Bottapotamon (Crustacea: Decapoda: Brachyura: Potamidae)"

_PeerJ, doi:10.7717/peerj.7980_

## Round 0.1 · original submission · Major Revisions

Both reviewers tended to be positive, but tidenitfied a number of issues that need to be addressed, particularly regarding a further exploration of your findings. Also, a good revision of the text would be highly desirable.

[]

·

Basic reporting

Many sections within the text are not clear, and are indicated in the text attached.

Many parts of the text are repeated in the different sections, without further explanations or new insights. Just repeat of the same text or paragraph.

The authors has provided a table containing the differences between the different species including the two species described but did not refer to the table much. The authors can discuss more on the morphological and adaptations, with inference to their molecular data and the variance event.

Experimental design

The authors should provide more details on the materials and methods:

a) Did they use old collections and newly collected materials? Or just new freshly collected materials for molecular analysis.

b) What and where were their comparative materials when establishing the two new species?

c) Whose protocol did they follow? What were the modifications to the original protocol etc.

Validity of the findings

The results are novel and interesting. However, the authors fail to address their findings in greater depth.

The authors could have also compared their results to other groups of organisms that can be be found within similar geographical ranges to see if these organisms have also experienced similar speciation event etc.

The authors provided figures 13 and 14, there are so much information that can be extracted from these two figures and very in-depth discussion that can be provided but not much has been written about it.

The results, discussions and conclusions are repeats of each other.

Additional comments

This is a very nice project and it is worth pursuing. A lot of information and speculations/hypotheses can be generated from this project.

·

Basic reporting

There are major problems with language throughout the text. I recommend the authors use an English editing service. There are also too many figures for a manuscript of such a nature, figs. 5 and 10 can be combined, figs. 11 and 12 can be combined, fig. 4 can be combined with another one. Other problems please see attached MS.

Experimental design

no additional comment, see attached MS for details.

Validity of the findings

no additional comment, see attached MS for details.

Additional comments

Please see attached MS for details and revise and make corrections throughout the text, then use a language editing service. I would be happy to review the MS again in detail after these have been done.

---

## Round 0.2 · Major Revisions

As indicated by the reviewers, there are several important issues that still were not addressed.

·

Basic reporting

The overall formatting requires a bit of work. The details are in the document attached.

Experimental design

No comment

Validity of the findings

No comment

Additional comments

Please see the details in the main text, and please be consistent in the formatting of the scientific names, when citing other people's article and in the references.

·

Basic reporting

The English used has been greatly improved upon using an editing service. However, there are still multiple errors in upper/lower case and wrong punctuation marks throughout the text. There is even a copied paragraph that is out of position. The references are still not in alphabetical order. The scientific names in the tables are not in italic etc… The amount of care into manuscript preparation is insufficient. The morphological description needs to be improved greatly, I suggest you use your teams' previous publications as a reference. See the attached file for details.

Experimental design

Ok.

Validity of the findings

I have some concerns about the measurements and ratios, please check your measuring methods and see if you have to redo all the measurements. See the attached file for details. If possible, use multiple specimens for the same measurement to make up for error. Some characters on your comparison table are highly variable, are you sure they are fit for comparison? If so, please give a brief justification because to my knowledge these characters have not been used before. Please see some recent publications on new Potamidae species to see which characters are useful.

Additional comments

The number of authors has changed from three to four. I am strongly against this (see: Ng, P. K. L. (1994). The citation of species names and the role of the authors name. Raffles Bulletin of Zoology, 42(3), 509-513.). As a referee, I am happy spend as much time as is needed in correcting scientific errors and give advice in basic reporting, but it is not our duty to tidy up a careless mistakes throughout the manuscript, please take care in preparing the manuscript and revise again and again before submitting. I am willing to revise once again after you have responded to all of my suggestions.

---

## Round 0.3 · Major Revisions

I was disappointed to see so many unaddressed issues, particularly those indicated by reviewer 2. Unless they are all carefully and properly incorporated into the text, I will not send it back to reviewers for another round of evaluation.

·

Basic reporting

The comments are listed in red inside the manuscript.

Experimental design

no comment

Validity of the findings

The authors need to explain more on figure 11.

Additional comments

Please see the comments inside the manuscript.

·

Basic reporting

The manuscript is improved, but there are still numerous issues that have not been addressed. For example: In line 293, "The male sternum is relatively flat and has granularwith numerous small pits;. sternites 1/2 fused and triangular; transverse sulcus between sternites 2/3 suture; sternites 3/4 fused without obvious demarcation. " is repeated, this has still not been deleted. In line 306, "third maxilliped without flagellum", but your description and photos show there is a flagellum. Also, you have to be more precise with the comparisons, which characters are only seen in the new species? Which species is the new species closest to and how do they differ? These additional details will be of much use to future researchers.

Experimental design

Same as last review. You MUST check your measurements, it is highly unlikey that the carapace ratio is 1.6-1.7, that is very very wide!

Validity of the findings

Findings are generally valid though the articulation needs to be improved.

Additional comments

Please address all issues I have previously pointed out.

---

## Round 0.4 · Minor Revisions

I believe that you properly addressed the remaining issues. However, there are still a number of typos throughout the manuscript that need to be fixed before the paper can be published. Please address these typos before we can finally Accept the manuscript.

---

## Round 0.5 · accepted · Accept

I believe the authors addressed all of the suggestions by the reviewers.